# Single Ventricle—A Comprehensive Review

**DOI:** 10.3390/children8060441

**Published:** 2021-05-24

**Authors:** P. Syamasundar Rao

**Affiliations:** McGovern Medical School, University of Texas-Houston, Children’s Memorial Hermann Hospital, 6410 Fannin Street, UTPB Suite # 425, Houston, TX 77030, USA; P.Syamasundar.Rao@uth.tmc.edu or srao.patnana@yahoo.com; Tel.: +1-713-500-5738; Fax: +1-713-500-5751

**Keywords:** single ventricle, double-inlet left ventricle, hypoplastic left heart syndrome, tricuspid atresia, unbalanced atrioventricular septal defect, mitral atresia with normal aortic root, heterotaxy syndromes, Down syndrome, Fontan circulation, biventricular repair

## Abstract

In this paper, the author enumerates cardiac defects with a functionally single ventricle, summarizes single ventricle physiology, presents a summary of management strategies to address the single ventricle defects, goes over the steps of staged total cavo-pulmonary connection, cites the prevalence of inter-stage mortality, names the causes of inter-stage mortality, discusses strategies to address the inter-stage mortality, reviews post-Fontan issues, and introduces alternative approaches to Fontan circulation.

## 1. Introduction

The term “single ventricle” is generally utilized to describe any congenital heart defect (CHD) with one functioning ventricle, and these are: double-inlet left ventricle (DILV), single ventricle, common ventricle, and univentricular atrio-ventricular (AV) connection [1]. Other lesions, namely hypoplastic left heart syndrome (HLHS), tricuspid atresia, unbalanced AV septal defect, mitral atresia with normal aortic root, and heterotaxy syndromes with one functioning ventricle may now be added to this group. The objectives of this paper are to specify congenital heart defects with one functioning ventricle, describe single ventricle physiology, present management strategies to address the single ventricle defects, cite the prevalence of and name the causes of inter-stage mortality, discuss strategies to address the inter-stage mortality, review post-Fontan issues, and to introduce alternative approaches to Fontan palliation.

## 2. Cardiac Defects with Functionally Single Ventricle

As mentioned above, there are number of cardiac defects that have a functionally single ventricle and are candidates for single ventricle/Fontan repair. These will be described briefly. The order of presentation is arbitrary.

### 2.1. Hypoplastic Left Heart Syndrome

The phrase “HLHS” was first suggested by Noonan and Nadas [2] to characterize a very small left ventricle with poorly developed aortic and mitral valves. The left ventricle (LV) is usually a slit-like cavity (Figure 1) with thick muscle, especially when mitral atresia co-exists. The aortic valve is atretic or markedly narrowed with annular hypoplasia. Similarly, the mitral valve is markedly stenotic, hypoplastic or atretic (Figure 1). The LV is very small when the mitral valve is open. Endocardial fibroelastosis is often present. Hypoplasia of the ascending aorta is present, and its diameter is usually 2–3 mm (Figure 2). However, it is adequate to supply ample coronary blood flow retrogradely. The left atrium is very small (Figure 1). The interatrial septum is usually thickened with a small patent foramen ovale (PFO) and rarely the atrial septum is intact. A patent ductus arteriosus (PDA) is classically present and is necessary for the baby to survive [3,4,5].

### 2.2. Tricuspid Atresia

Tricuspid atresia (TA) is a cyanotic CHD and is characterized as a congenital absence or agenesis of the morphologic tricuspid valve [6,7]. The right atrium is dilated and the tricuspid valve is atretic (Figure 3). In the most common muscular variety, atretic tricuspid valve is seen as a localized fibrous thickening or a dimple in the floor of the right atrium at the anticipated site of the tricuspid valve [8,9,10,11,12]. An atrial septal defect is necessary for survival and is typically a stretched PFO. The mitral valve is usually bicuspid and morphologically a mitral valve. The LV is evidently a morphological LV, but it is enlarged and hypertrophied [8,9,10,11,12]. Usually, a ventricular septal defect (VSD) is present. The VSD is most commonly in the muscular ventricular septum [13,14]. The right ventricle (RV) is hypoplastic and is not sufficiently large in size to support pulmonary circulation. The origin of great arteries is variable and on the basis of which a classification of this disease entity was developed [6]: Type I, normally related great arteries; Type II, d-transposition of the great arteries; Type III, malpositions of the great arteries other than d-transposition; and Type IV, truncus arteriosus [6]. Some of the echocardiographic features are demonstrated in Figure 3.

### 2.3. Double-Inlet Left Ventricle

In DILV, the ventricle most frequently has LV morphology, although RV, mixed, indeterminate, or undifferentiated morphologies have been reported [1,15]. The main ventricle is largely a morphological LV with an outlet chamber connected to it which has a RV morphology. The AV valves may be normal (Figure 4), or one of them may be hypoplastic, stenotic, or even atretic. The great arteries are most frequently transposed and the aorta arises from the hypoplastic RV, and the pulmonary artery comes off the main LV chamber. L-transposition happens more often than d-transposition. The great vessels are normally related in 30% of cases. Double-outlet RV may be seen, in which both great vessels arise from the rudimentary RV. Pulmonary stenosis is present in two thirds of patients and such stenosis is seen irrespective of the great artery relationship. The stenosis may be at the valvar or subvalvar level, or the pulmonary valve/artery may be atretic. Subaortic obstruction may be present in patients with transposition of the great arteries and is due to the stenosis of the VSD or, more accurately, bulbo-ventricular foramen. Patients with subaortic obstruction often have associated aortic coarctation [1,15].

### 2.4. Unbalanced Atrioventricular Septal Defect

Defects in the atrial and ventricular septae, together with defects in one or both AV valves, described in the past as common AV canal or complete endocardial cushion defect [16], are now generally called AV septal defects (AVSDs). In the complete form, there is usually a large inlet VSD contiguous with a primum atrial septal defect (ASD), and a common AV valve with a single valve annulus. A variant of this, called the intermediate form, has the same features but with distinct and separate right and left AV valves. Transitional and partial forms are also described, and are similar to or the same as ostium primum ASDs. Complete AVSDs are also categorized based on the ventricular sizes, namely balanced and unbalanced (Figure 5) [17]. Unbalanced AVSDs comprise 10–15% of all complete AVSDs. The unbalanced types may be LV-dominant with a large LV and small RV, or RV-dominant with a large RV and small LV (Figure 5). RV-dominant AVSDs are more frequently seen [17].

### 2.5. Mitral Atresia with Normal Aortic Root

Mitral atresia may be seen with aortic atresia or a normal aortic root; the former is generally characterized as HLHS which was discussed in a preceding section. In this segment, mitral atresia with normal aortic root will be reviewed. While this division is arbitrary, it is considered appropriate [18] because of the differences in therapy for these two defects, in that HLHS babies almost always require the Norwood procedure [19,20] in the neonatal period, while babies with mitral atresia with normal aortic root need different types of intervention at different ages based on the associated cardiac defects [18]. Mitral atresia with normal aortic root is an uncommon complex CHD and is usually associated with several other defects. The atretic mitral valve may be an absent AV connection or an imperforate valve membrane. The left atrium (LA) is hypoplastic or small (Figure 6) and the right atrium is enlarged and hypertrophied. A PFO or ASD is generally seen; a PFO in 2/3rds of patients and an ASD in 1/3rd. A restrictive atrial septal defect may be present and, rarely, atrial septum is intact; in such cases the levoatriocardinal vein may help to decompress the LA. A single ventricle is most commonly associated ventricular anatomy, although in a few cases there are two ventricles. If two ventricles are present, the LV is small and communicates with the RV via a VSD which is usually small (Figure 6). The RV is always enlarged and hypertrophied. Transposition of the great vessels is frequently seen. Double-outlet right ventricle is present in some babies. In most patients the pulmonary valve is normal and not stenotic. However, in 25–30% of cases valvar and/or subvalvar stenosis or atresia is seen. The aortic valve and aortic root are near normal in size by definition. Coarctation of the aorta or interrupted aortic arch is seen in nearly 30% of cases. Such aortic obstruction is seen only in subjects with a normal pulmonary valve. PDA is seen in nearly 80% of patients [15,18].

### 2.6. Other Heart Defects with Single Ventricle Physiology

There are other cardiac defects that may have functionally one ventricle. These will be briefed.

#### 2.6.1. Pulmonary Atresia with Intact Ventricular Septum

Some children with pulmonary atresia with intact ventricular septum have severe hypoplasia of the RV, infudibular atresia or RV-dependant coronary circulation and become candidates for single ventricle repair [21,22,23,24]. Similarly, patients who end-up with very small RVs despite transcatheter and/or surgical procedures to improve the RV size will also join this group [21,22,23,24].

#### 2.6.2. Ebstein’s Anomaly of the Tricuspid Valve

Some severe forms of Ebstein’s anomaly of the tricuspid valve have extremely small RVs, unable to support pulmonary circulation, are also candidates for single ventricle repair [25,26,27].

#### 2.6.3. Heterotaxy Syndromes

Some patients with asplenia and polysplenia syndromes have only one functioning ventricle [28,29] and become candidates for single ventricle repair consideration.

## 3. Single Ventricle Physiology

In normal subjects the circulation is in series (Figure 7); the vena caval blood is emptied into the right atrium (RA), and from there is passed on to the RV, and pulmonary artery (PA) and is transported into the lungs for oxygenation. From there, it is returned to the LA, LV, and aorta and is passed on to the body for delivery of oxygen and nutrients. The resultant systemic arterial saturation is ≥96%. The blood is returned back to the vena cavae and the cycle is repeated (Figure 7).

By contrast, in babies with single ventricle, the systemic (SBF) and pulmonary (PBF) blood flow returns mix with each other in the single functioning ventricle (Figure 8) with consequent lower systemic oxygen saturations [30]. The single ventricle then provides both the SBF and PBF (Figure 8). This admixture in the single ventricle causes systemic arterial oxygen desaturation. The O_2_ saturations vary between 75% and 85%, depending upon the quantity of PBF and pulmonary to systemic flow ratio (Qp:Qs).

The PBF is either derived from the pulmonary artery (PA) coming off of the single ventricle or from the level of the great vessels (patent ductus arteriosus, surgically placed aorta-pulmonary artery shunt (or RV to PA non-valved conduit), or aorto-pulmonary collateral vessels). The SBF is either unobstructed or obstructed.

Based on such a thought, these patients may be grouped into the following subsets [30]; this classification is helpful in determining the type of therapy that a given patient needs.
Group 1: Unobstructed Systemic Blood Flow

1A: Unobstructed SBF with unobstructed PBF

1B: Unobstructed SBF with moderately obstructed PBF

1C: Unobstructed SBF with severely obstructed (or atretic) PBF


Group 2: Obstructed Systemic Blood Flow


In summary, the systemic and pulmonary blood flow returns mix with each other with consequent lower systemic oxygen saturations (75–85%—Normal ≥96%). The systemic and pulmonary circuits work in-parallel rather than normal in-series circulation. Markedly increased PBF results in systemic hypo-perfusion and severely decreased PBF creates severe hypoxemia. Therefore, a delicate balance of the blood flows between the two circulations should be preserved.

## 4. Management Strategies to Address the Single Ventricle Defects

Since there is only one functioning ventricle, the overall objective is to allow the functioning ventricle (whether morphologically right or left) to supply the systemic circulation and to connect the systemic veins directly to the PAs (Fontan circulation—single ventricle repair). This procedure can’t be done in the neonate since PA pressure and resistance are high. Consequently, it is performed by staged procedures. The types of procedures and their timing have evolved over the years, as reviewed elsewhere [10,31,32,33,34], since their original description by Fontan, Kruetzer and their associates [35,36]. Currently, this is accomplished by staged total cavo-pulmonary connection (TCPC), devised by de Leval and associates [37]. This TCPC is undertaken in three stages: Stages I, II, and III.

### 4.1. Stage I—Management at the Time of Initial Presentation

The management at initial presentation, typically in the neonatal period, depends upon the pathophysiology of the defect itself plus the associated cardiac abnormalities, namely the status of PBF, the presence of obstruction to SBF, and other cardiac defects.

#### 4.1.1. Status of Pulmonary Blood Flow

The PBF may be decreased, increased, or the PBF is adequate.

##### Reduced Pulmonary Blood Flow (Group 1C above)

In patients with reduced PBF, the ductus arteriosus should be kept patent by intravenous infusion of prostaglandin E_1_ (PGE_1_) at a dose of 0.05–0.1 mcg/kg/min. After the O_2_ saturation gets better, the dose of PGE_1_ is slowly, stepwise, decreased to 0.02–0.025 mcg/kg/min to lessen the adverse effects of the prostaglandins. Subsequent to achieving a stable baby and studies needed to firm up the diagnosis, a more stable way of supplying the PBF should be undertaken. Several methods of augmenting PBF have been utilized in the past, as reviewed elsewhere [38]. Of these, modified Blalock–Taussig (BT) shunt [39], ductal stenting [22,40,41], balloon valvuloplasty of the pulmonary valve (if the major obstruction is at the level of pulmonary valve) [42,43,44], and most recently connection of the RV outflow tract with the PA via a non-valved Gore-Tex tube [45] are currently available options. However, most surgeons use a modified BT shunt by placing of a Gore-Tex graft between the subclavian artery and the PA on the same side [39] (Figure 9) to address pulmonary oligemia.

##### Increased Pulmonary Blood Flow (Group 1A above)

A markedly elevated PBF produces congestive heart failure (CHF). These infants are initially stabilized with anticongestive therapy [46]. This should be followed by banding of the PA [47] (Figure 10), irrespective of control of CHF.

##### Adequate Pulmonary Blood Flow (Group 1B above)

Babies with a mildly increased or near normal PBF with O_2_ saturations in the low 80s do not exhibit significant symptoms and, are less cyanotic than babies with pulmonary oligemia and do not require any treatment and may be followed until Stage II.

#### 4.1.2. Obstructed Systemic Blood Flow (Group 2 above)

The majority of these babies belong to HLHS syndrome or its variants and require Norwood operation [19,20] with either Modified BT [39] or Sano [48,49] shunt (Figure 11). In some institutions, cardiac transplantation [50] is employed to address this lesion.

#### 4.1.3. Other Cardiac Abnormalities

Some of the babies may have other cardiac abnormalities such as intra-cardiac obstruction, aortic arch obstruction, or atrioventricular valve insufficiency at presentation or may develop such abnormalities during the succeeding period. These will be reviewed.

##### Intra-Cardiac Obstruction

Intra-cardiac obstruction may take place either at the atrial or at the ventricular level.

##### Inter-Atrial Obstruction

The total systemic or pulmonary venous return must pass through the PFO in babies with atretic atrioventricular valves and/or atretic semi-lunar valves. These defects are listed in Table 1.

There is a tendency for the PFO to remain open because of a persistence of fetal flow patterns. However, sometimes the PFO becomes obstructed. Evidence for obstructed PFO may be present either with systemic or pulmonary venous congestion and is confirmed by a small-sized PFO by 2-dimentional echo and high velocity flow across it by Doppler studies. The PFO may be enlarged by balloon atrial septostomy (Figure 12) [52]. Such a procedure is usually successful, especially in the early neonatal period, because the septum primum (lower margin of the PFO) is thin and flail and can be ruptured by balloon septostomy.

When PFO obstruction develops during the later part of the neonatal period or in older infants, balloon atrial septostomy is unlikely to be successful, and in such situations, alternative methods such as blade atrial septostomy [53], static balloon dilatation [54,55], or stent placement [56,57,58] may have to be used to accomplish the relief of inter-atrial obstruction. If transcatheter methods are not feasible or not successful, surgical atrial septectomy is necessary. If there is no PFO and the atrial septum is intact, the atrial septum may be perforated either by Brockenbrough technique [59] or radiofrequency perforation [60]. This should be followed by static dilatation [54,55] or stent placement across the atrial septum [56,57,58]. For a detailed discussion of these transcatheter techniques, the interested reader may review the author’s previous publications [58,61].

In babies with single ventricle and mitral atresia, inter-atrial obstruction is frequently present [62,63]. In such infants, predictable fall in pulmonary vascular resistance (PVR) takes place after relief of atrial obstruction either by balloon septostomy or surgery [63]. Therefore, banding of the PA should be performed without hesitation at the time of alleviating the atrial septal obstruction in order to reduce the probability for CHF, reduce the PVR and PA pressure, prevent pulmonary vascular obstructive disease (PVOD), and pave the way for Fontan circulation [62,63].

##### Inter-Ventricular Obstruction

In some complex cardiac defects, namely, tricuspid atresia, double-inlet left ventricle, and double-outlet right (or left) ventricle, the VSD or bulbo-ventricular foramen, as the case may be, is very small and obstructive at presentation in the neonatal period or spontaneously becomes smaller with time [13,14,64,65]. Such a reduction in the size of the VSD or bulbo-ventricular foramen causes sub-pulmonary obstruction, producing reduced pulmonary blood flow or subaortic narrowing, resulting in obstruction to systemic blood flow (Table 2).

If such a closure produces diminished pulmonary blood flow, the management is similar to that discussed in the “Decreased Pulmonary Blood Flow” section above (PGE_1_ and modified BT shunt). If the VSD/bulbo-ventricular foramen closure results in obstruction to SBF, the narrowing is either relieved directly by enlarging the VSD or the obstruction is bypassed by the Damus–Kaye–Stansel (DKS) procedure (anastomosis of the divided pulmonary artery to the ascending aorta either directly or via a prosthetic conduct) (Figure 13) [14,66]. Most commonly DKS is performed instead of directly enlarging the VSD. However, it should be noted that the development of a significant inter-ventricular obstruction that requires intervention in the neonate is infrequent.

##### Aortic Obstruction

Aortic arch obstruction may occur in the form of aortic coarctation or interrupted aortic arch. Both will be briefly reviewed.

##### Aortic Coarctation

Aortic coarctation is a descending aortic abnormality with constriction of the aortic arch distal to the left subclavian artery at about the level of ductal insertion [67]. In the neonate, it is frequently associated with other defects such as PDA, VSD and other complex CHDs such as tricuspid atresia with transposition of the great arteries and DILV with transposition of the great arteries. Initial medical management by infusion of PGE_1_ to bypass the coarctation and anti-congestive treatment, if CHF is present, is suggested. In most institutions the neonatal coarctations are addressed by surgery. Although the immediate success rate following balloon angioplasty is good [68,69,70], because of a high recurrence rate [70,71,72] associated with balloon therapy in the neonate, surgery is preferred. However, in special circumstances when surgical risk is high or contraindicated, balloon dilatation may be used as an initial treatment option [68,69,70,73].

##### Interrupted Aortic Arch

In aortic arch interruption, there is a total loss of luminal continuity between the arch of the aorta and the descending aorta [74] and the condition is classified into types A, B, and C [75] depending upon the site of interruption. The initial management is by PGE_1_ infusion to allow the ductus to open and restore systemic perfusion. Then, end-to-end anastomosis after removal of the ductal tissue is performed. In cases where the aortic arch can’t be mobilized, an interposition Gore-Tex graft is inserted to bridge the gap. In any baby with interrupted aortic arch, especially in association with single ventricle physiology, arch repair should be promptly performed.

##### Atrio-Ventricular Valve Insufficiency

Mild AV valve insufficiency does not require any treatment. However, moderate to severe AV valve insufficiency (tricuspid, mitral or common AV valve) should be addressed promptly. Initially medical management with angiotensin-converting-enzyme (ACE) inhibitors (Captopril/Enalopril) may be used. If no adequate improvement with medical therapy is seen, surgical valvuloplasty or valve replacement may become necessary.

### 4.2. Stage II—Bidirectional Glenn

Independent of the nature of palliative intervention during the neonatal period, bidirectional Glenn operation [76] by anastomosis of the superior vena cava (SVC) to the right PA, end-to-side (Figure 14) is undertaken at an approximate age of six months. If a prior BT or Sano shunt is present, it is ligated at the time of bidirectional Glenn. While performing the procedure at six months is commonly accepted, the Glenn can be performed as early as three months subject to demonstrating normal PA pressures and anatomy.

In babies who have persistent left SVC, bilateral bidirectional Glenn (Figure 15) is performed particularly in patients with a small or absent left innominate vein. A bidirectional Glenn procedure may also be undertaken for patients with infrahepatic interruption of the IVC with azygos or hemiazygos continuation, and such a procedure may be called the Kawashima procedure.

Prior to undertaking the bidirectional Glenn procedure, it must be ensured that the PA pressures are normal and the branch PAs are adequate in size. This is most often accomplished by cardiac catheterization and cineangiography. However, some institutions employ echocardiogrms or other imaging studies such as magnetic resonance imaging (MRI) or computed tomography (CT) to accomplish this goal. If stenosis of the PAs is present, it may be relieved with balloon angioplasty or stent placement, as appropriate, or it may be repaired while performing the bidirectional Glenn procedure. If atrioventricular valve regurgitation, aortic coarctation, subaortic obstruction, and other abnormalities are present, they should also be repaired at the time of bidirectional Glenn surgery.

### 4.3. Stage III—Fontan/Kruetzer Procedure

Although the author prefers the term “Fontan/Kruetzer Procedure” because of the simultaneous description of the procedure by both groups [35,36], it is more commonly referred to as “Fontan Procedure” in the literature and therefore, it will be so used in the rest of the presentation. During the final stage, the IVC blood flow is rerouted into the PA. At the same time a fenestration between the conduit and the atrial mass is created. Arbitrary division of these procedures into Stage IIIA (diversion of IVC into the PA) and Stage IIIB (closure of the fenestration) [33,34] may be undertaken.

#### 4.3.1. Stage IIIA

In the Stage IIIA, the TCPC may be accomplished by redirecting the IVC blood flow into the PA either by a lateral tunnel [77,78] or by an extra-cardiac, non-valved conduit (Figure 16) [79,80]. This surgery is typically carried out anytime from one to two years of age, frequently one year after the bidirectional Glenn. At the present time, the majority of surgeons favor an extra-cardiac conduit to achieve the final stage of Fontan. It also appears that most surgeons create a fenestration, 4–6 mm in size, between the conduit and the atria (Figure 16) [81]. Whereas creation of fenestration during the Fontan surgery was originally suggested for patients with high-risk [81,82], most surgeons and pediatric intensivists appear to opt for fenestration, because the creation of fenestration during the Fontan decreases mortality rate and lessens the morbidity during the immediate postoperative period [33].

#### 4.3.2. Stage IIIB

During the Stage IIIB, the fenestration is occluded (Figure 17) by transcatheter methods [33,81,83,84,85], usually 6–12 months following Stage IIIA Fontan. In the past, all previously available ASD occluding devices [81,83,84,85] were used for fenestration closure. However, at the present time, Amplatzer Septal Occluders are the most regularly used devices to accomplish fenestration closures. Any other residual shunts may also be addressed by device closure.

## 5. Inter-Stage Mortality

As mentioned above, in babies who have single ventricle physiology, the systemic and pulmonary circulations function in-parallel instead of the usual in-series circulation and a delicate equilibrium between the two circulations must be maintained such that sufficient systemic and pulmonary perfusions are preserved. Inability to maintain such balance may result in morbidity and even mortality in these vulnerable babies. A substantial inter-stage mortality, ranging from 5% to 15%, has been documented [33,34,86,87]. Some investigators have identified the reason for inter-stage mortality; these are restrictive atrial septal defect, obstructed aortic arch, distorted pulmonary arteries, AV valve regurgitation, shunt blockage, and inter-current illnesses [86,88]. The inter-stage mortality is seen more often between Stages I and II than between Stages II and III.

Strategies to address the inter-stage mortality include periodic clinical evaluation, as well as echo-Doppler and other imaging studies to detect the above described abnormalities and provide adequate relief of detected problems in order to prevent/reduce the mortality.

### 5.1. Restrictive Atrial Communication

Obstruction of PFO/ASD may manifest either as systemic venous or pulmonary venous congestion, depending upon the lesion. The obstruction may be confirmed by a small-sized PFO by 2-dimentional echo and high velocity flow across it by Doppler. Once it is detected, it should be promptly relieved. In the majority of babies, Rashkind balloon atrial septostomy [52,58,61] (Figure 12) is successful in relieving the interatrial obstruction. Rashkind septostomy may not be feasible in some patients either because of thick atrial septal tissue and/or small left atria. In such situations, atrial septal restriction may be relieved by static balloon dilatation [54,55] (Figure 18) or stent implantation [55,56,57,58,61,89] (Figure 19 and Figure 20). If transcatheter methods are not feasible or not successful, surgical septostomy becomes necessary.

### 5.2. Obstruction of the Aortic Arch

Aortic coarctation may develop or a missed neonatal diagnosis may manifest subsequent to Stage I palliation, or aortic recoarctation may occur following prior surgery. These may be detected by physical examination (decreased and/or delayed femoral arterial pulses and higher systolic blood pressure in the arm than leg) or imaging studies such as echocardiogram or MRI/CT. The indications for intervention are CHF or hypertension along with a peak systolic gradient higher than 20 mmHg. At this stage, balloon angioplasty [69,70,72,90] (Figure 21) may be carried out.

Good results may be expected with both native [67,68,69,70,72,73] (Figure 22 and Figure 23) and post-surgical aortic recoarctations [89,90,91,92,93,94] (Figure 24 and Figure 25).

However, if there is long-segment coarctation or associated transverse aortic arch hypoplasia, surgical therapy to address the aortic obstruction should be considered.

### 5.3. Distortion/Stenosis of the Pulmonary Arteries

If distortion or stenosis of the branch PAs is detected, balloon angioplasty for discrete obstruction and stents [95,96,97,98,99] for long segment or diffuse narrowing may be indicated (Figure 26). Again, surgery is reserved for situations that can’t be addressed by transcatheter approaches.

### 5.4. Atrio-Ventricular Valve Insufficiency

Atrio-ventricular valve insufficiency may be addressed by administration of afterload reducing agents (ACE inhibitors {Captopril/Enalopril}), surgical valvuloplasty or valve replacement, as appropriate.

### 5.5. Shunt Blockage

Some of the babies with one functioning ventricle may have had palliation with a modified BT [39] or a Sano [48,49] shunt. Both these shunts may become occluded either completely or partially, producing acute or chronic hypoxemia, respectively. If such an obstruction develops before a planned bidirectional Glenn procedure in single-ventricle physiology patients, further surgery such as a shunt revision may be needed, further increasing the risk of inter-stage mortality and/or morbidity. In almost all of these babies the shunt is the sole source of PBF. Therefore, prompt assessment, diagnosis, and treatment are critical to guarantee a successful result.

Babies with a completely obstructed shunt manifest with marked cyanosis and respiratory distress. A continuous murmur of the shunt that was present previously is no longer heard on auscultation or is markedly decreased in intensity. The management of an obstructed shunt includes use of principles of basic life support, namely, the airway, breathing, and circulation (or circulation, airway, breathing as per the new protocol) with intubation, as necessary, and starting cardiopulmonary resuscitation. In babies with suspected shunt blockage, heparin should be administered to prevent futher progression of the thrombus. Concurrently, stat echocardiogram should be performed, and pediatric cardiology and cardiovascular surgical consultation should be obtained promptly. Transcatheter recanalization of the shunt (Figure 27) in an urgent manner is feasible by interventional pediatric cardiologists [58,87,100] or the shunt may be revised, or the baby placed on ECMO by pediatric cardiovascular surgeons, depending upon the baby’s clinical status and institutional preference. The shunt thrombosis appears to occur, though infrequently, despite the routine use of platelet inhibiting drugs such as aspirin in an attempt to avert such a problem.

Stenotic shunts without acute symptoms may undergo electives balloon angioplasty or stent implantation across the stenotic BT or Sano shunts (Figure 28 and Figure 29).

As discussed above, timely medical, trans-catheter or surgical treatment as appropriate should be undertaken in an attempt to prevent mortality and reduce morbidity.

### 5.6. Inter-Current Illnesses

Inter-current illnesses that produce dehydration, acidosis, or high fever may disturb the equilibrium between the pulmonary and systemic circulations and the infant may become critically ill [86,87]. Indeed, these inter-current illnesses result in significant inter-stage mortality [86]. Consequently, even minor illnesses must be addressed promptly. Any baby with high temperature, diarrhea, vomiting, or reduced fluid intake should be watched until the fever subsides for at least 24 h. IV fluids should be administered until the vomiting, or diarrhea resolves, or until oral intake normalizes. Therefore, intense attention in managing these patients should be continued by the caregiver [86,87]. Even minor illnesses must be aggressively monitored and addressed as deemed appropriate.

## 6. Post-Fontan Issues

Following completion of Fontan, periodic follow-up is necessary, namely assessment at one, six, and twelve months following Stage IIIB, and once a year thereafter is suggested. Inotropic and/or diuretic therapy is provided as deemed appropriate. Reduction of afterload with an angiotensin-converting enzyme inhibitor (Captopril or Enalopril) is instituted by most cardiologists. Anticoagulation with platelet-inhibiting doses of aspirin (2–5 mg/kg/day) in infants and children or clopidogrel (75 mg/day) in adults to prevent thrombo-embolism is a routine for most patients.

The results of older types of Fontan (RA-to-PA or RA-to-RV anastomosis either directly or via valved or non-valved conduits) indicated high initial mortality rates varying from 10% to 26% [31,32,101,102]. In addition, the postoperative stay in the intensive care unit (ICU) was long. The initial mortality following staged TCPC without fenestration decreased remarkably with rates ranging from 8% to 10.5% [103,104,105]. There was a further reduction in mortality rates to 4.5% to 7.5% when TCPC with fenestration was employed [106,107,108].

A number of complications were detected during follow-up: arrhythmias, obstructed Fontan pathways, persistent shunts, thrombo-embolism, development of cerebro-vascular accidents (CVA), cyanosis, systemic venous to pulmonary venous collateral vessels, and systemic venous congestion including protein-losing enteropathy [10,33,109]. The complications seem to occur more often with earlier types of Fontan than with the current staged TCPC with extra-cardiac conduit and fenestration. When such complications are detected, they should be quickly investigated and treatment provided as detailed elsewhere [33,34,109].

## 7. Alternative Approaches to Fontan Circulation

Despite improvement in mortality rates with staged TCPC, there is a significant morbidity with Fontan. Poor outcomes are seen in children with Down syndrome, heterotaxy syndromes, young age at surgery, high mean PA pressure (>15 mmHg), AV valve insufficiency, a morphologic tricuspid valve as systemic AV valve, distorted PAs, poor ventricular function, non-sinus rhythm, presence of a pacemaker, an atrio-pulmonary type of Fontan, no Fontan fenestration, and long cardiopulmonary bypass time [107,110,111,112,113,114,115]. Therefore, alternative approaches to Fontan are being entertained and these include, one and one-half ventricle repair, primary biventricular repair, staged biventricular repair, and conversion from single-ventricle (SV) to two-ventricle (TV) repair.

### 7.1. One and One-Half Ventricle Repair

In one and one-half ventricle repair, a bidirectional Glenn procedure is undertaken and allows the small RV to handle the reduced volume. The ASD/PFO is closed. Babies with pulmonary atresia with intact ventricular septum whose RVs have not grown adequately to support pulmonary circulation are candidates for this approach. In extreme forms of Ebstein’s anomaly with a minimal functional right ventricle, bidirectional Glenn anastomosis may help to improve the hemodynamic outcome. Anecdotal experiences show favorable results, but no organized studies or long-term results are available at this time.

### 7.2. Primary Biventricular Repair

In primary biventricular repair, the AV valves are reconstructed so that nearly similar-sized AV valves are produced and the ASD is closed. The VSD is closed with a patch to create nearly equal-sized ventricles; the patch is relocated to the right in RV-dominant AVSDs and to the left in the LV-dominant AVSDs. Inflow and outflow obstructions, if present, are relieved. and endocardial fibroelastosis resected.

The criteria used for selection of patients for primary biventricular repair are not clearly established. A number of echocardiographic, MRI, and cine-angiographic criteria have been examined as reviewed elsewhere [17,116]. A combination of LV Z scores and LV volumes seems to help decide on such a selection.

### 7.3. Staged Biventricular Repair

In staged biventricular repair, the ASD and VSD are partly closed at the time of the initial operation, relief of inflow and outflow obstruction, if present, is provided; endocardial fibroelastosis, if present, is resected; and additional Glenn/BT shunt are created to promote blood flow into the left atrium and ventricle. The ASD and VSD are completely closed at the time of a second surgery after the growth of the hypoplastic ventricle is verified. Any other residual abnormalities are also taken care of at the time of second operation.

### 7.4. Results of Biventricular Repair

The data of the results of biventricular repair are limited. The mortality rates ranged between 10.4% and 18% [117,118,119]. Survival at long-term was excellent and varied between 88% and 90% [117,118,120]. However, the requirement for operative (17.4–34%) and transcatheter (13.4%) re-interventions was high [117,118,120]. Nevertheless, an increase in AV valve Z-scores (−2.8 to −7.4 vs. −0.6 to −2.7) [118], ventricular size Z-scores (−1.0 to −7.5 vs. −2.0 to +1.8) [118], and LV end-diastolic volume Z-scores (from a median of −3.15 to +0.42) [117] was seen at follow-up.

### 7.5. Conversion from Single-Ventricle to Two-Ventricle Repair

During conversion from SV to TV repair, relief of inflow and outflow tract obstructions and restriction of the atrial defect to promote flow through the LV were undertaken in an attempt to rehabilitate the LV. If endocardial fibroelastosis is present, it is also resected.

The mortality after conversion from SV to TV is low (1–11%) [117,121] However, operative and catheter re-interventions were needed during follow-up in 19% and 38% patients respectively [121]. While the initial LV Z-scores before Stage I SV palliation were similar, the Z-scores of the LV increased significantly in the SV to TV conversion group and the Z scores decreased in the SV palliation group who have not had SV to TV conversion [122]. Improvement of LV end-diastolic volume by echo from 28.1 to 58.5 mL/m^2^ was also demonstrated in a limited number of the unbalanced AVSD patients after SV to TV conversion [123].

### 7.6. Comparison of Various Methods of Intervention

When mid-term results of primary, staged biventricular repair, and SV to TV conversion of unbalanced complete AVSDs were examined, there was low morbidity and mortality, but surgical and/or catheter re-interventions were required in 52% of patients [124]. Nathan and her associates evaluated the results of patient subsets who underwent SV palliation, biventricular repair, and SV to TV conversion [117]. The biventricular repair and SV to TV conversion cohorts had similar mortality and the necessity for heart transplantation but these rates were lower than those seen with the SV palliation cohort at a mean follow-up of 35 months [117]. While the need for operative re-interventions was similar in the three cohorts, the need for catheter re-interventions was lower in the biventricular repair cohort than in the other two cohorts. These authors conclude that biventricular repair and SV to TV conversion may be performed with relatively lower mortality and morbidity rates in children with unbalanced AVSDs. SV to TV conversion and biventricular recruitment appear to be a vital option for children with Down syndrome and heterotaxy syndrome because of anatomic and physiologic reasons. Babies with Down syndrome tolerate SV physiology poorly [115]. Children with heterotaxy syndrome have intricate and abnormal AV valve anatomy and require complicated and innovative repair methods. These patients tolerate AV valve insufficiency poorly when SV palliation is opted for.

## 8. Other Issues

### 8.1. Role of Socioeconomic Status, Racial Classification, and Geographical Location in the Management of Single Ventricle Patients

The author has practiced pediatric cardiology in five academic centers for the last 50 years and has not discovered significant adverse influence of these factors in the management of CHD patients in general and single ventricle patients in particular. The sole exception is a baby with HLHS whose parents declined to care for their baby (secondary to socioeconomic issues), but the hospital’s Social Service Department rapidly identified adoptive parents so that appropriate care could be provided by the medical staff. However, how much the lack of access to medical care by some of these groups [125] will influence the pediatric cardiac care has not been thoroughly investigated.

### 8.2. Is There a Need for Centralization of Surgical Care of Complex CHD

The available data, though sparse, indicate a slightly better surgical outcomes in larger-volume institutions than in smaller-volume programs. Requiring patients to be transferred to selected centers will impose substantial burden to the parents in addition to medical insurance coverage issues. Whether such an approach is desirable is at best marginal, given the minor better outcome advantage at large-volume institutions.

### 8.3. Hybrid Approach for HLHS

In HLHS patients with high risk for Norwood surgery, particularly in premature babies, a hybrid approach has been utilized. In the first stage of this procedure, banding of both branch pulmonary arteries through median sternotomy and stent implantation into the ductus arteriosus via pulmonary arteriotomy (Figure 30 and Figure 31), originally described by Akintuerk and associates [126], are performed simultaneously [126,127,128,129]. This becomes Stage I.

During stage II, aortic arch reconstruction, atrial septectomy, and a bidirectional Glenn shunt are performed. This is followed by a Fontan conversion (Stage III). It would appear that there is a reduction of early mortality when this approach is used, but some of it is transferred to Stage II. A careful comparison of the hybrid approach with the conventional Norwood procedure did not reveal an improvement in outcome following the hybrid procedure [128]. In addition, a lower systemic and cerebral oxygen delivery with the hybrid procedure than with the Norwood procedure [129] is of concern. It appears that the majority of institutions are reverting to a conventional Norwood procedure with a Sano shunt. Our own personal preference is to reserve the hybrid procedure for premature babies with HLHS and those that have other co-morbidities that preclude safe Norwood palliation.

### 8.4. Cardiac Transplantation

Transplantation of the heart is another available surgical alternative [50] for patients with HLHS and other complex CHD in the neonate. Following the report of Bailey and associates [50], several institutions have applied this procedure to address HLHS. However, because of scarcity of donor hearts, most institutions have reverted back to Norwood. At the present time, cardiac transplantation is employed in patients with failed Fontan. However, since these patients, having been exposed to cardiopulmonary bypass and blood products, may develop HLA antibodies, which increases their risk of antibody mediated rejection following transplantation.

## 9. Summary and Conclusions

Several cardiac defects have functionally single ventricle; notable of these are: hypoplastic left heart syndrome, tricuspid atresia, single ventricle (DILV), unbalanced AV septal defect, mitral atresia, and others. Single ventricle physiology is dissimilar from that of biventricular physiology and a fragile balance of the blood flows between the pulmonary and systemic circulations should be preserved. These babies are generally managed by staged total cavo-pulmonary connection (Fontan) in three stages. The prevalence of inter-stage mortality is high (5–15%). It is more frequent between Stages I and II than between Stages II and III. The inter-stage mortality is largely associated with: restrictive atrial communication, obstruction of the aortic arch, distortion/stenosis of the pulmonary arteries, AV valve insufficiency, shunt blockage, and inter-current illnesses. Periodic clinical evaluation and echocardiographic or other imaging studies to detect these abnormalities should be systematically undertaken and when identified, they should be promptly treated. Even minor inter-current illnesses must be addressed aggressively.

Post-Fontan complications decreased remarkably following the introduction of staged TCPC with extracardiac conduit and fenestration with succeeding catheter-based fenestration occlusion. Because of the problems identified with single ventricle (Fontan) repair, alternative methods, namely one and one-half ventricle repair, primary biventricular repair, staged biventricular repair, and conversion from SV to TV repair, are being considered.

## Figures and Tables

**Figure 1 children-08-00441-f001:**
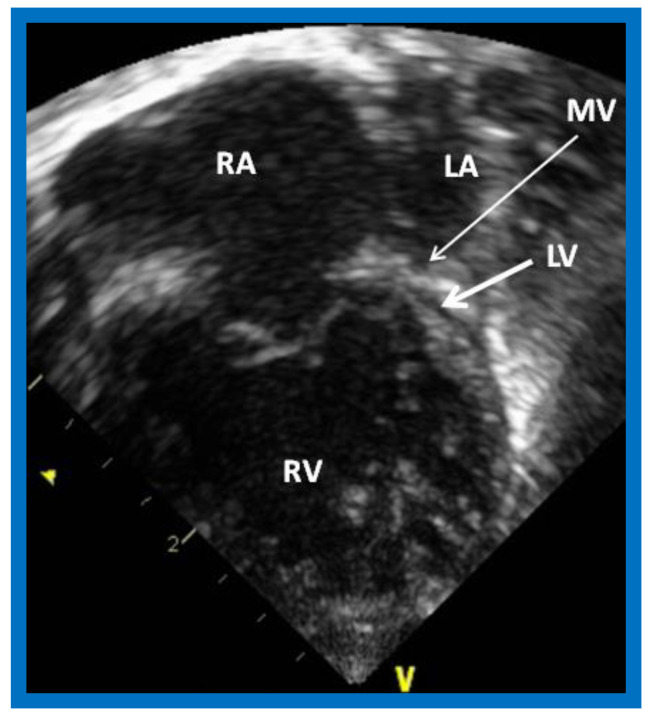
Echocardiogram in an apical 4-chamber view of a baby with hypoplastic left heart syndrome, showing a strikingly small, slit-like (thick arrow) left ventricle (LV), obviously enlarged and hypertrophied right ventricle (RV), and a dilated right atrium (RA). The atretic mitral valve (MV) (thin arrow) and hypoplastic left atrium (LA) are also seen.

**Figure 2 children-08-00441-f002:**
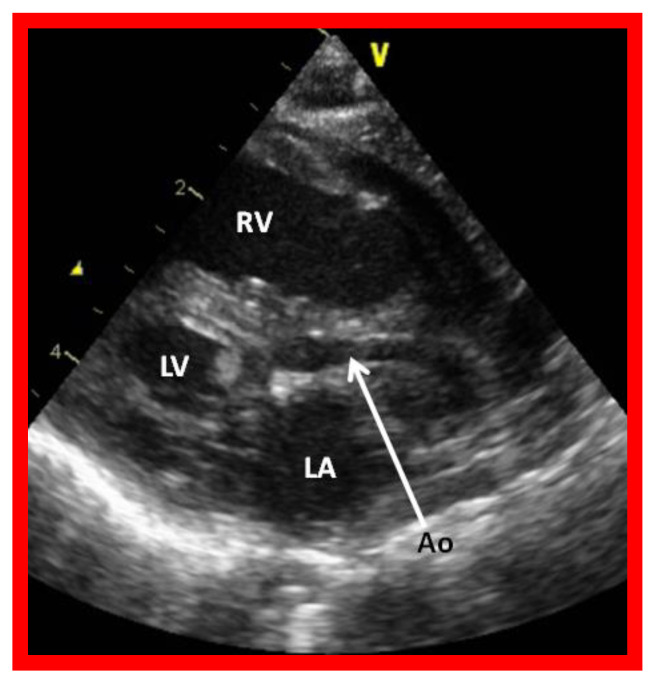
Echocardiogram in a parasternal long short axis view of a baby with hypoplastic left heart syndrome illustrates a small left ventricle (LV), a severely hypoplastic aorta (Ao) (arrow), and an enlarged right ventricle (RV). LA, left atrium.

**Figure 3 children-08-00441-f003:**
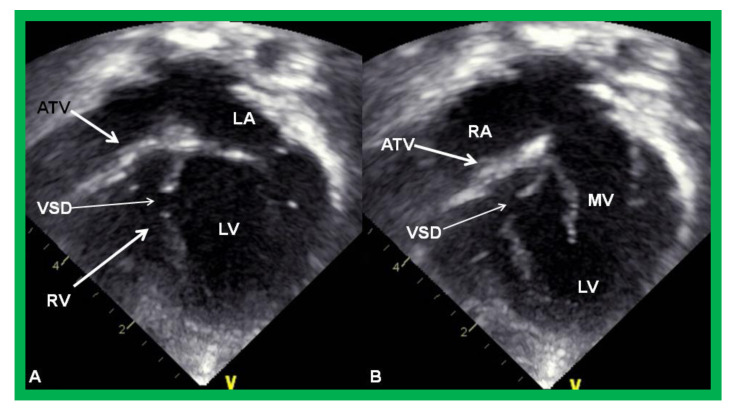
Echocardiograms in apical four-chamber views of an infant with tricuspid atresia demonstrating a dilated left ventricle (LV), a small right ventricle (RV), and a dense band of echoes at the site where the tricuspid valve echo should be (ATV; thick arrow). Images with closed (**A**) and open (**B**) mitral valve are shown; the tricuspid valve remains closed in both situations. A ventricular septal defect (VSD; thin arrow) is also shown. LA, Left atrium; RA, Right atrium.

**Figure 4 children-08-00441-f004:**
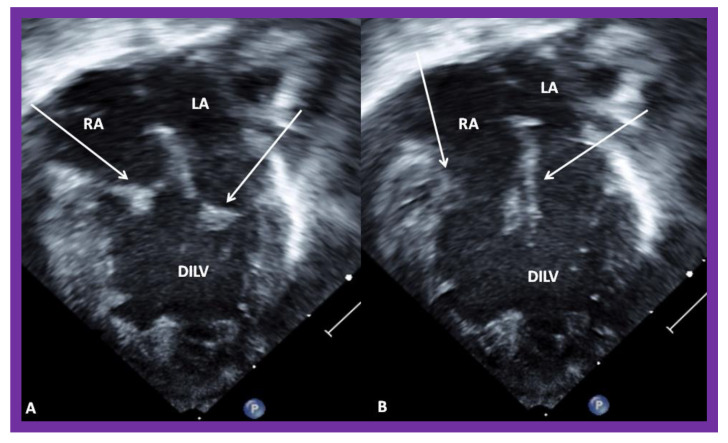
Echocardiograms in apical four-chamber views of a baby with double inlet left ventricle (DILV) with closed (**A**) and open (**B**) atrioventricular valves (arrows). The outlet chamber is not visualized in this view. LA, left atrium; RA, right atrium. Reproduced from Reference [15].

**Figure 5 children-08-00441-f005:**
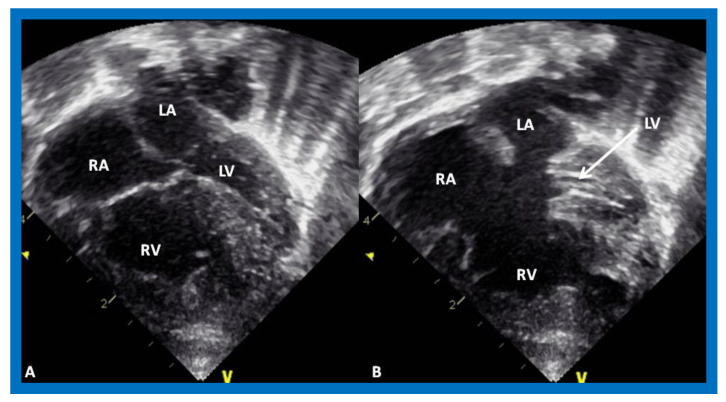
Echocardiograms in apical four-chamber views of an infant with complete atrioventricular septal defect with marked hypoplasia of the left ventricle (LV). LV in diastole (**A**) and in systole (**B**) are shown. LA, left atrium; RA, right atrium; RV, right ventricle.

**Figure 6 children-08-00441-f006:**
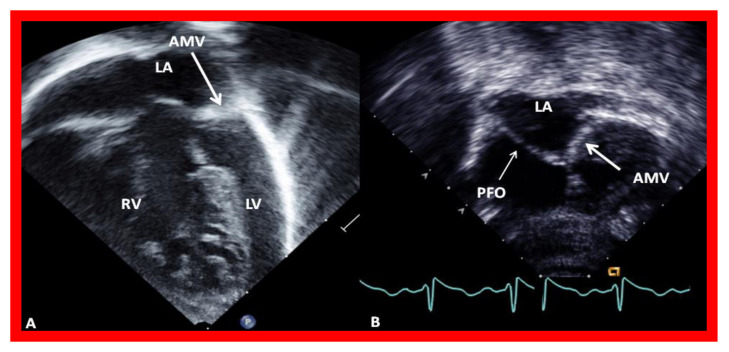
Echocardiograms in modified four-chamber views of two infants with mitral atresia, demonstrating atretic mitral valves (AMV), indicated by thick arrows. A small left atrium (LA) and left ventricle (LV) and a large right ventricle (RV) are also seen. The thin arrow in (**B**) shows a restrictive patent foramen ovale (PFO). All four chambers were shown in (**A**) while (**B**) focuses on the atria. Reproduced from Reference [15].

**Figure 7 children-08-00441-f007:**
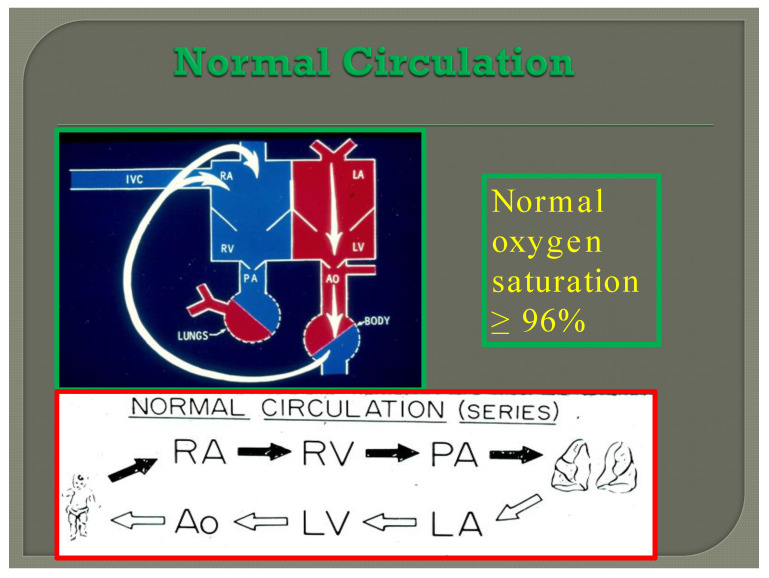
Diagram illustrating normal circulation in series (see the text) resulting systemic arterial oxygen saturation of ≥96%.

**Figure 8 children-08-00441-f008:**
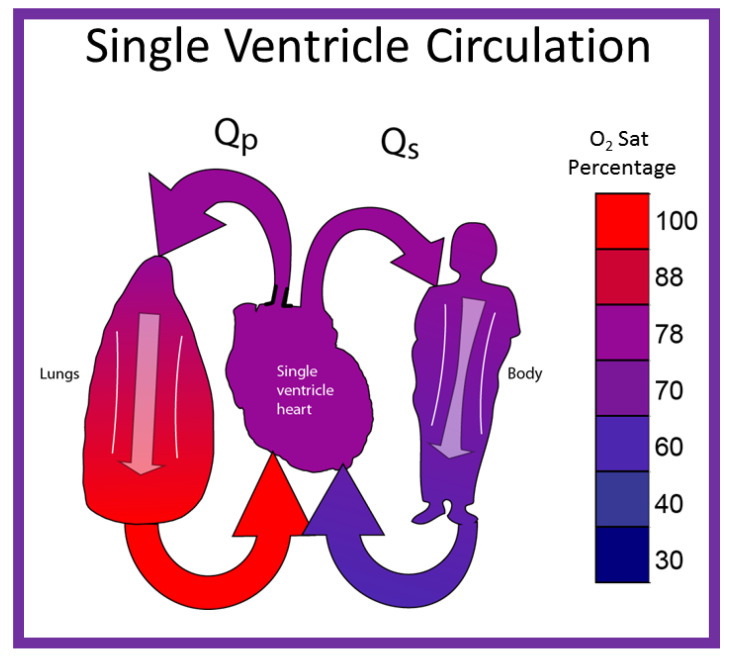
Diagrammatic portrayal of single ventricle circulation; the blood from both the lungs and body return to the single ventricle; this mixed blood is redistributed to the lungs (Qp) and body (Qs). Reproduced with permission from Reference [30].

**Figure 9 children-08-00441-f009:**
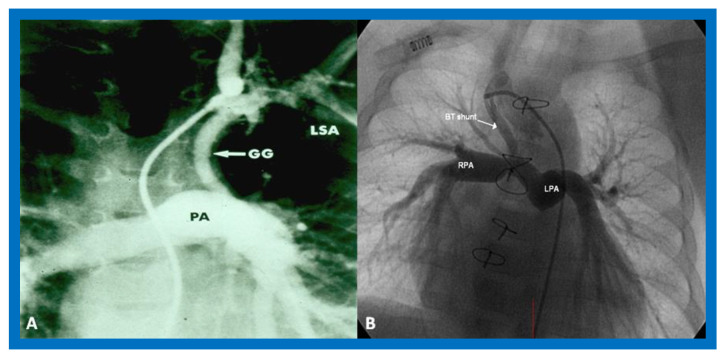
Cine-angiograms demonstrating patent Gore-Tex grafts (GG) following modified Blalock-Taussig (BT) shunt surgery in two patients. Note wide-open BT shunts with good visualization of the pulmonary artery (PA) in (**A**) of the right (RPA) and left (LPA) pulmonary arteries in (**B**).

**Figure 10 children-08-00441-f010:**
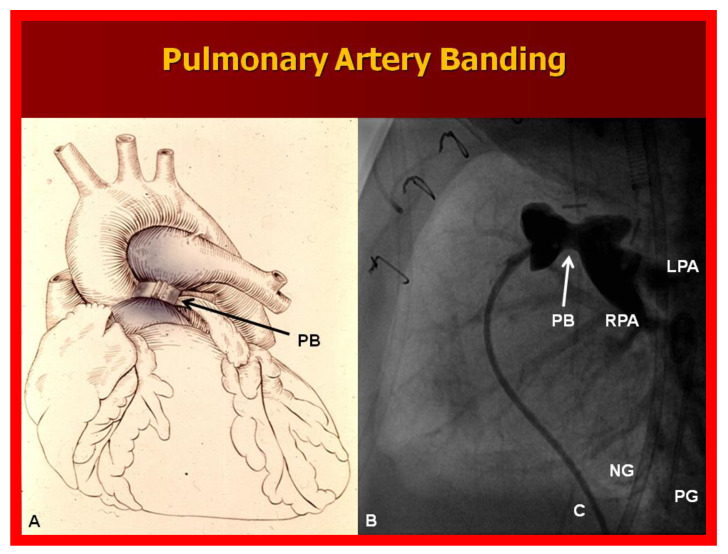
(**A**) Diagrammatic portrayal of pulmonary artery banding (PB) for infants with markedly elevated pulmonary blood flow and congestive heart failure. (**B**) Selected cine frame form pulmonary artery cineangiogram in straight lateral view demonstrating constriction of the pulmonary artery (PB; arrow) in an infant who had the banding procedure. C, catheter; LPA, left pulmonary artery; NG, nasogastric tube; PG, pigtail catheter; RPA, right pulmonary artery.

**Figure 11 children-08-00441-f011:**
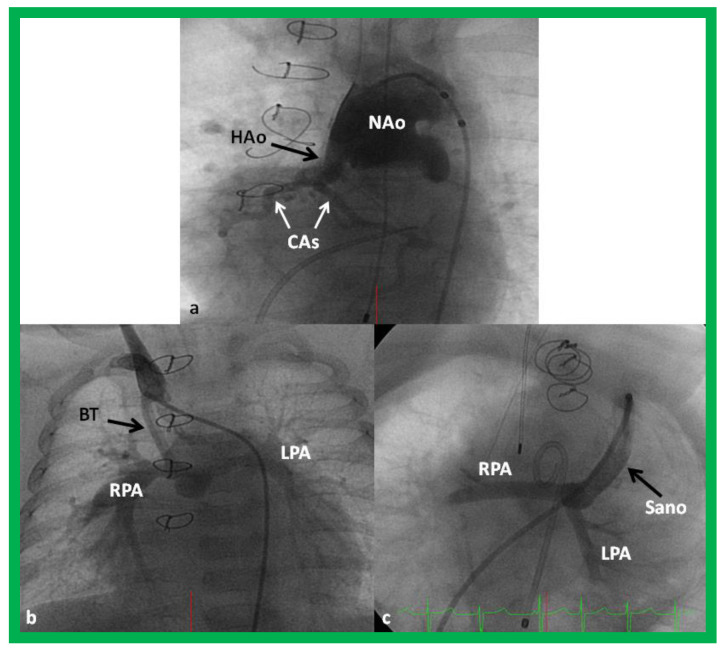
Cineangiogrames demonstrating the Norwood procedure, showing the neoaorta (NAo) and hypoplastic old aorta (HAo). The latter supplies the coronary arteries (CAs) as shown in (**a**). A Blalock-Taussig (BT) shunt in seen in (**b**) and a Sano shunt in (**c**) from two other babies are also shown. This is Stage I of the Fontan procedure for infants with hypoplastic left heart syndrome. LPA, left pulmonary artery; RPA, right pulmonary artery. Reproduced from Reference [33].

**Figure 12 children-08-00441-f012:**
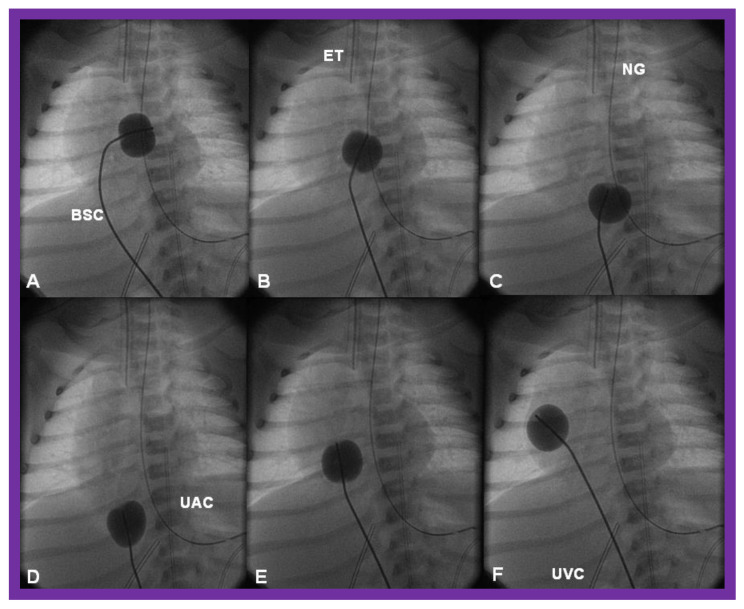
Cinfluroscopic frames demonstrating the procedure of Rashkind’s balloon septostomy. Initially the balloon is inflated in the left atrium (**A**). The balloon septostomy catheter (BSC) is rapidly and forcefully pulled into the right atrium (**B**) and inferior vena cava (**C**,**D**) and quickly advanced back into the right atrium (**E**,**F**). The entire procedure is done as one single motion. Rapid advancement of the BSC into the right atrium (**E**,**F**) is done in order to avoid inadvertent occlusion of the inferior vena cava if failure to deflate the balloon occurs (this is quite rare). At about the same time the balloon is deflated. ET, endotracheal tube; NG, nasogastric tube; UAC, umbilical arterial catheter; UVC, umbilical venous catheter.

**Figure 13 children-08-00441-f013:**
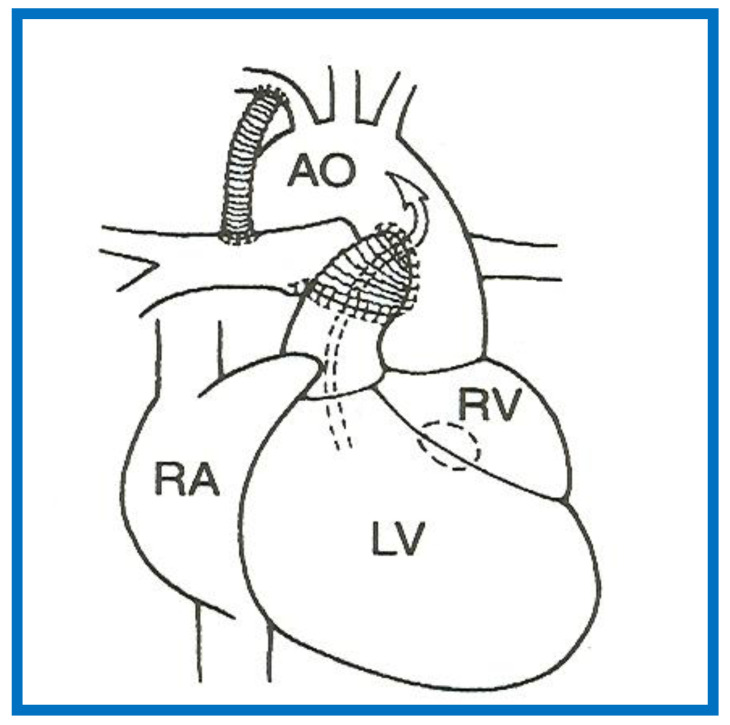
A Line drawing illustrating Damus-Kaye-Stansel procedure. The left ventricular (LV) blood flows via the ventricular septal defect (circle) and right ventricle (RV) into the aorta (Ao). If the VSD is small and restrictive, causing “subaortic” obstruction, this obstruction may be circumvented by connecting the proximal stump of the divided pulmonary artery to the Ao directly or a via a non-valved conduit. The pulmonary arteries are supplied with a Blalock-Taussig shunt. Concept derived from References [13,14].

**Figure 14 children-08-00441-f014:**
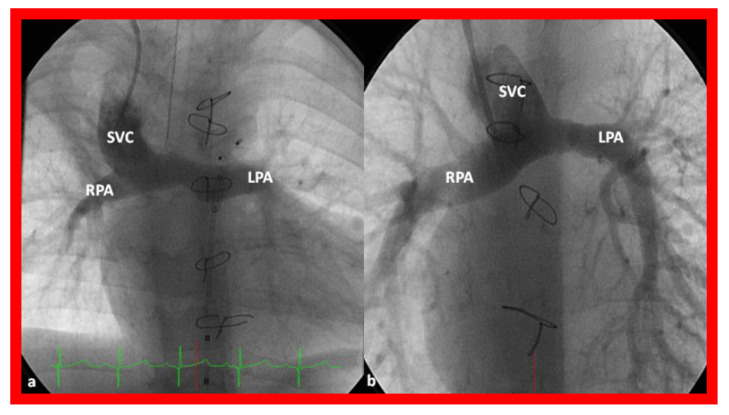
Cineangiographic frames illustrating bidirectional Glenn procedure, i.e., anastomosis of the superior vena cava (SVC) to the right pulmonary artery (RPA)] in two different children is shown in (**a**,**b**) (Stage II). Unobstructed blood flow from the SVC to the right (RPA) and left (LPA) pulmonary arteries is seen. Reproduced from Reference [33].

**Figure 15 children-08-00441-f015:**
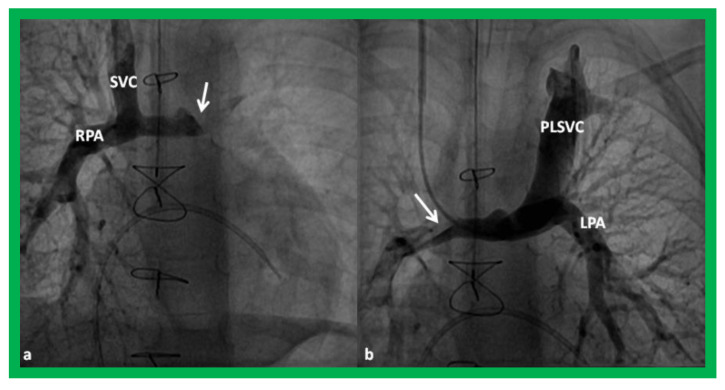
Cineangiographic frames demonstrating a bilateral bidirectional Glenn procedure (Stage II). In (**a**), angiogram of the superior vena cava (SVC) illustrates opacification of the right pulmonary artery (RPA). The arrow in (**a**) shows the unopacified blood from a persistent left superior vena cava (PLSVC). In (**b**), an injection into the PLSVC illustrates opacification of the left pulmonary artery (LPA). The arrow in (**b**) shows the unopacified blood from the right SVC. Unobstructed flow from the respective SVCs into the pulmonary arteries is clearly seen. Reproduced from Reference [33].

**Figure 16 children-08-00441-f016:**
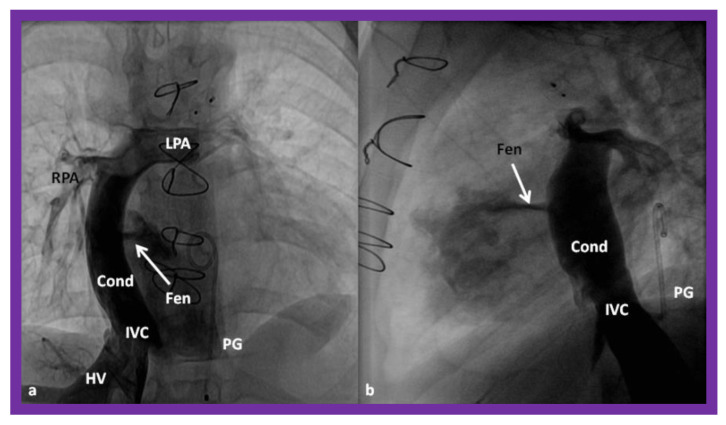
Cineangiographic frames in postero-anterior (**a**) and lateral (**b**) projections, illustrating Stage IIIA of the Fontan procedure in which the inferior vena caval (IVC) blood flow is diverted into the pulmonary arteries via a non-valve conduit (Cond). The flow via the fenestration (Fen) is shown by the arrows in a and b. HV, hepatic veins; LPA, left pulmonary artery; PG, pigtail catheter in the descending aorta; RPA, right pulmonary artery. Modified from Reference [33].

**Figure 17 children-08-00441-f017:**
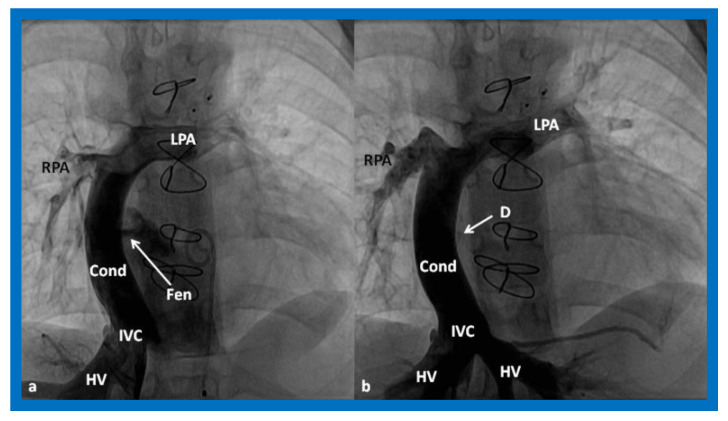
(**a**)**.** Cineangiogram in antero-posterior view, illustrating Stage IIIA of the Fontan operation, diverting the inferior vena caval (IVC) blood flow into the pulmonary arteries via a non-valve conduit (Cond). Fenestration (Fen) is shown by the arrow in (**a**). The Fen is occluded with an Amplatzer device (D), shown by the arrow in (**b**) (Stage IIIB). HV, hepatic veins; LPA, left pulmonary artery; RPA, right pulmonary artery. Reproduced from Reference [33].

**Figure 18 children-08-00441-f018:**
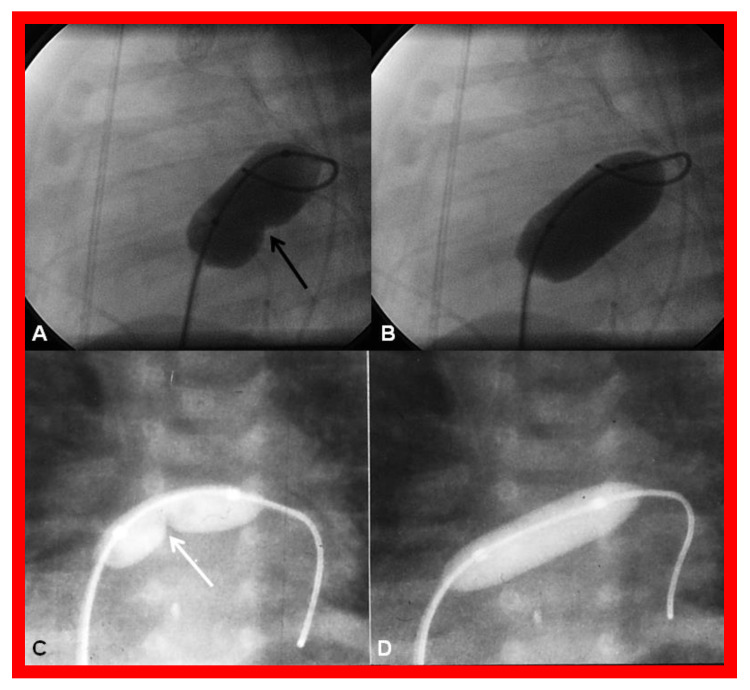
Cineradiographic frames demonstrating balloon angioplasty procedures to widen the patent foramen ovale. Inflated balloons in lateral (**A**,**B**) and posterior-anterior (**C**,**D**) projections in two different neonates, illustrating the waisting of the balloons (arrows in **A**,**C**) during the initial phases of balloon angioplasty. The waisting was fully abolished on further inflation of the balloons (**B**,**D**). Reproduced from Reference [89].

**Figure 19 children-08-00441-f019:**
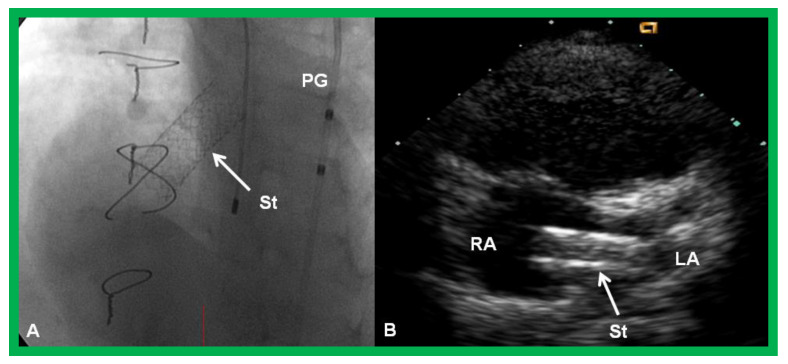
Selected cine (**A**) and video (**B**) frames demonstrating the location of the stent (St) across the atrial septum following St deployment. LA, left atrium; RA, right atrium; PG, pigtail catheter in the descending aorta. Reproduced from Reference [89].

**Figure 20 children-08-00441-f020:**
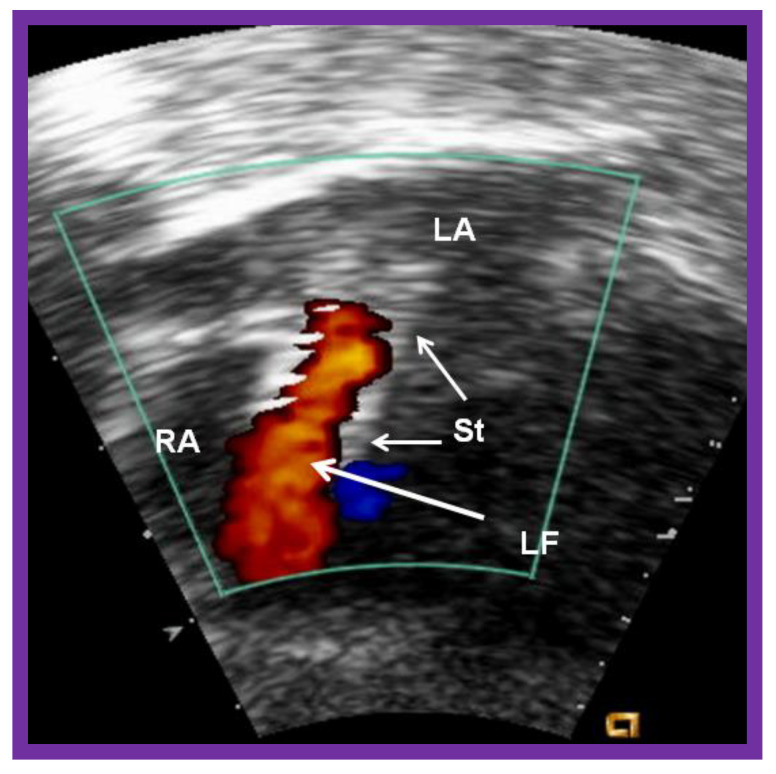
Video frame of the stent (St) (short arrows) demonstrates laminar flow (LF) (long arrow) across it. LA, left atrium; RA, right atrium. Reproduced from Reference [89].

**Figure 21 children-08-00441-f021:**
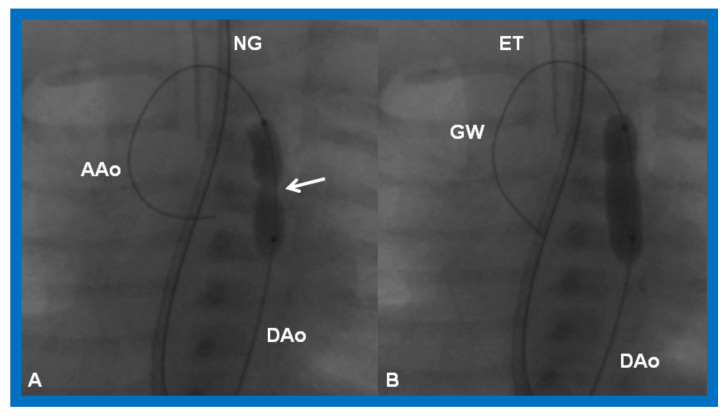
Cineradiographic frames in 20° left anterior oblique view demonstrating a balloon angioplasty catheter placed across the aortic coarctation. Note waisting (arrow) of the balloon (**A**) during the initial phases of balloon inflation, which was abolished (**B**) on further inflation of the balloon. AAo, ascending aorta; DAo, descending aorta; ET, endotracheal tube; GW, guide wire; NG, nasogastric tube. Modified from Reference [90].

**Figure 22 children-08-00441-f022:**
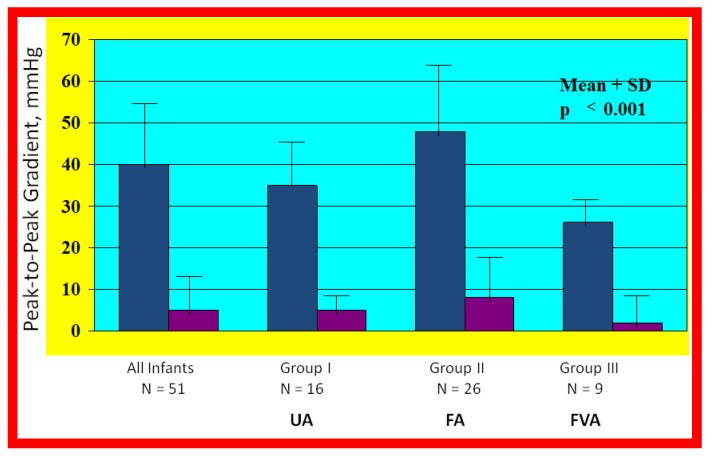
Bar graph illustrating the fall (*p* < 0.001) of the peak-to peak systolic pressure gradients (in mmHg) across the aortic coarctation following balloon angioplasty. The reduction in the gradients was seen for all the infant group (left panel) and for all the three subgroups: Balloon angioplasty via trans-umbilical arterial (UA), trans-femoral arterial (FA), and trans-femoral venous anterograde (FVA) routes. The mean and standard deviation (SD) are marked. N represents the number of subjects in each group. Modified from Reference [70].

**Figure 23 children-08-00441-f023:**
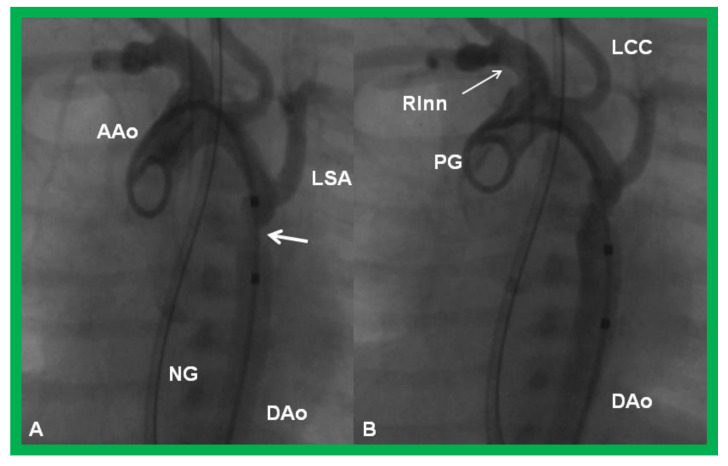
Cineangiographic frames from aortic arch angiograms in a 20° left anterior oblique views, illustrating a narrowed (coarcted) aortic segment (arrow) priot to balloon angioplasty (**A**) which improved following balloon angioplasty (**B**). Note mild hypoplasia of the distal transverse aortic arch and isthmus. AAo, ascending aorta; DAo, descending aorta; LCC, left common carotid artery; LSA, left subclavian artery; NG, nasogastric tube; PG, pigtail catheter; RInn. right innominate artery. Modified from Reference [90].

**Figure 24 children-08-00441-f024:**
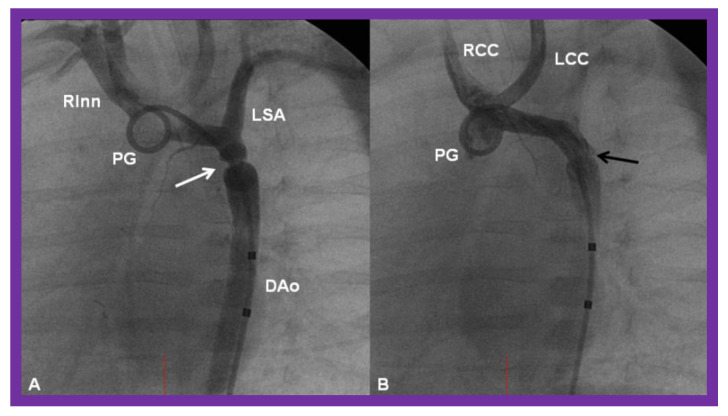
Cineangiographic frames from aortic arch angiograms in 20° left anterior oblique projection, illustrating a coarcted aortic segment (white arrow) prior to balloon angioplasty (**A**) which widened (black arrow) following balloon angioplasty (**B**) in a neonate who had had the Norwood procedure earlier. DAo, descending aorta; LCC, left common carotid artery; LSA, left subclavian artery; PG, pigtail catheter; RCC, right common carotid artery; RInn, right innominate artery. Reproduced from Reference [58].

**Figure 25 children-08-00441-f025:**
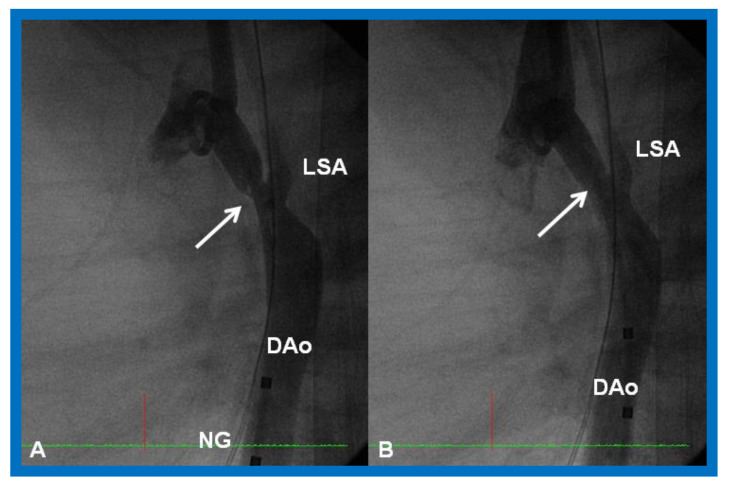
Cineangiographic frames from aortic arch angiograms in straight lateral projection showing a post-surgical recoarcted aortic segment (arrow) priot to balloon angioplasty (**A**) which improved (arrow) following balloon dilatation (**B**), in a neonate who developed recoarctation at three weeks of age after neonatal surgical repair of coarctation. DAo, descending aorta; LSA, left subclavian artery; NG, nasogastric tube. Reproduced from Reference [58].

**Figure 26 children-08-00441-f026:**
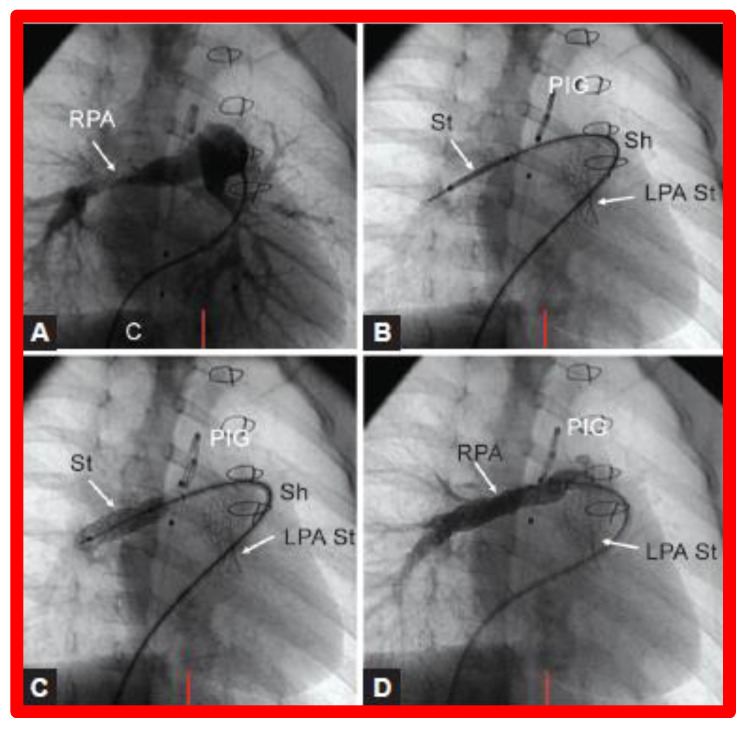
(**A**). Cineangiographic frames in a 30° right anterior oblique view illustrating long-segment right pulmonary artery (RPA) stenosis (arrow) before implantation of stent (St). The position of the St prior to (**B**) and following (**C**) balloon inflation to implant the St are shown. (**D**). Angiography following St implantation shows improvement. Note trivial residual narrowing (top arrow) in (**D**). C, catheter; LPA St, left pulmonary artery stent implanted just prior to RPA stent implantation. Sh, sheath; PIG, pigtail catheter. Reproduced from Reference [98].

**Figure 27 children-08-00441-f027:**
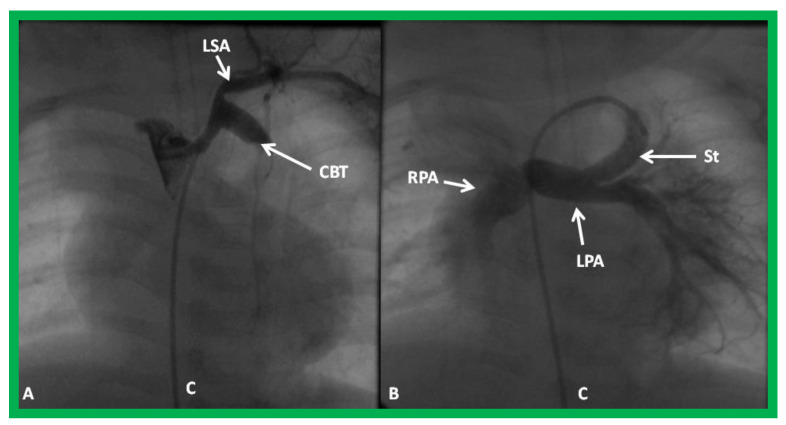
Selected cineangiographic frame demonstrating clotted Blalock-Taussig (CBT) shunt (**A**). After unsuccessful recanalization with mechanical thrombolysis and balloon angioplasty, a stent (St) was implanted (**B**). Angiography with tip of the catheter (C) in the proximal portion of the stent showed complete opening of the BT shunt with visualization of right (RPA) and left (LPA) pulmonary arteries. LSA, left subclavian artery. Reproduced from Reference [61].

**Figure 28 children-08-00441-f028:**
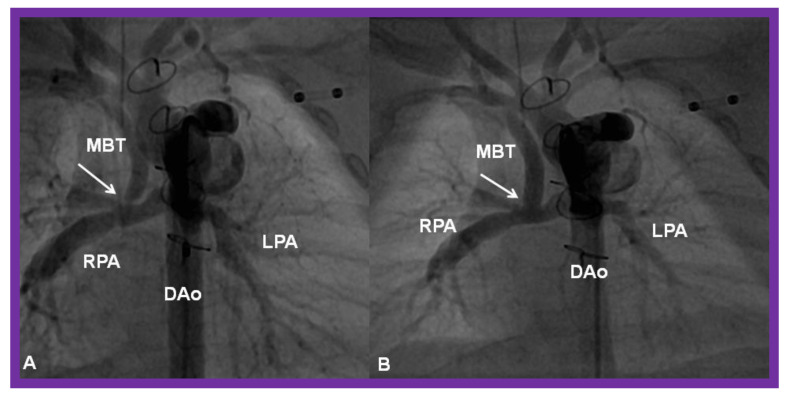
(**A**) Selected cine frame in a postero-anterior view, demonstrating discrete narrowing (arrow) of a modified Blalock-Taussig (MBT) shunt. (**B**). Following stent implantation, this site is wide open (arrow in (**B**)). DAo, descending aorta; LPA, left pulmonary artery; RPA, right pulmonary artery. Reproduced from Reference [58].

**Figure 29 children-08-00441-f029:**
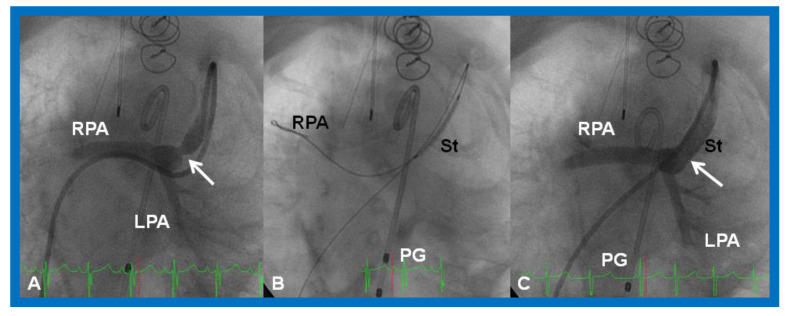
(**A**). Cineangiographic frame in a caudal angulation, demonstrating a narrowed (arrow) Sano shunt in a baby with hypoplastic left heart syndrome. (**B**). A stent (St) catheter is placed across the narrowed site with the guide wire positioned deep into the right pulmonary artery (RPA). (**C**). Note the wide-open (arrow) Sano shunt after the stent was implanted across the narrowed segment. LPA, left pulmonary artery; PG, pigtail catheter. Reproduced from Reference [58].

**Figure 30 children-08-00441-f030:**
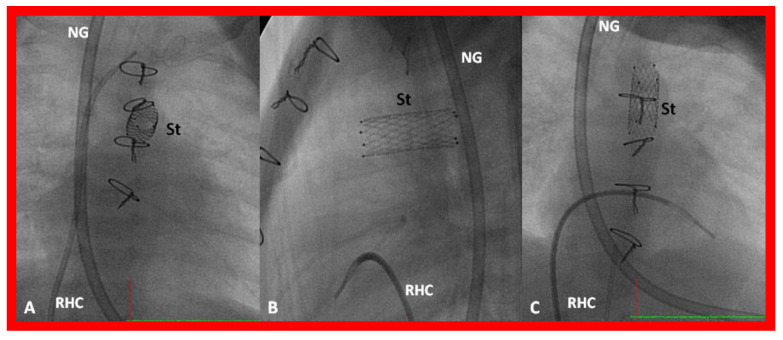
Selected cine frames in postero-anterior (**A**), lateral (**B**) and sitting-up (**C**) views, showing the position of a stent (St) placed within the ductus via a purse-string suture in the pulmonary artery in a premature neonate with hypoplastic left heart syndrome during a hybrid procedure. NG, naso-gastric tube; RHC, right heart catheter. Sternal wires are seen. Reproduced from Reference [58].

**Figure 31 children-08-00441-f031:**
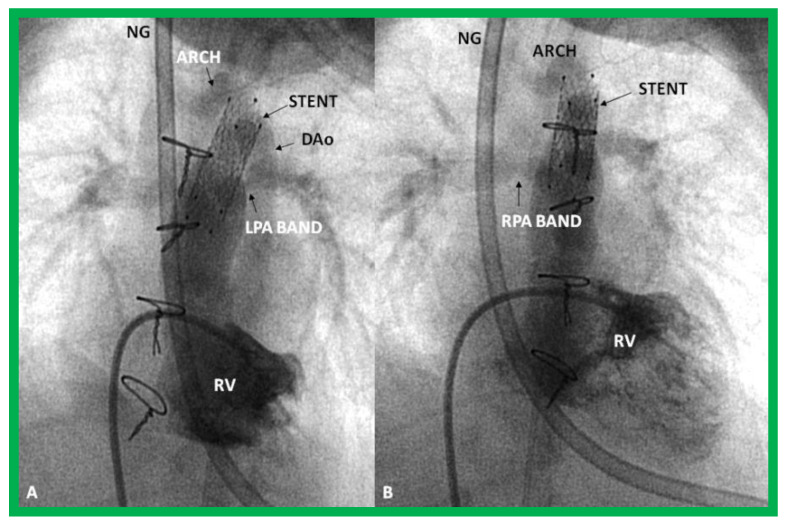
Selected cine frames in sitting-up (**A**) and right anterior oblique (**B**) views from a right ventricular (RV) angiogram, demonstrating the position of a stent (STENT) placed within the ductus via a purse-string suture in the pulmonary artery during a hybrid procedure, in a premature infant with hypoplastic left heart syndrome shown in Figure 30. Note the opacification of the descending aorta (DAo) via the stent. Bands placed during the hybrid procedure around the right (RPA) (in **B**) and left (LPA) (in **A**) pulmonary arteries are seen. ARCH, aortic arch opacified retrogradely via the STENT. NG, naso-gastric tube.

**Table 1 children-08-00441-t001:** Inter-atrial Obstruction.

1	Pulmonary atresia
2	Tricuspid atresia
3	Mitral atresia
4	Hypoplastic left heart syndrome

Modified from Reference [51].

**Table 2 children-08-00441-t002:** Inter-Ventricular Obstruction.

1	Tricuspid atresia
2	Double-inlet left ventricle
3	Double-outlet right (or left) ventricle

Modified from Reference [51].

## Data Availability

Not applicable.

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
