# Peer review of "Single Ventricle—A Comprehensive Review"

_children, 2021, doi:10.3390/children8060441_

Round 1
Reviewer 1 Report
Thank you for allowing me to review "Single Ventricle - A Comprehensive Review" by Dr Rao. This was a very well written review on single ventricle cardiac lesions. It is well organized, easy to read, and very comprehensive. The only modifications I would potentially have would be minor wording adjustments that would not drastically change the content of the manuscript and thus were not worth mentioning.
Author Response
I than the reviewer for the complimentary remarks. The manuscript was re-read and minor changes suggested were incorporated into the final script. Thanks again.
Reviewer 2 Report
The author presents an overview of single ventricle disease from physiology to management. The paper reads well and is well-supported by references and figures. I applaud the author for his comprehensive overview, and have only some comments:
- Can the author discuss the role and growing understanding of disparities (e.g., socioeconomic status, racial, geographical, etc.) in the management and outcomes of single ventricle patients?
- Given the complexity of single ventricle repair, outcomes and associated costs have widely varied across the United States. Centralization of such repairs in select, specialized centers may thus be needed. Can the author comment on the role of an institution’s/surgeon’s volume relative to their outcome (i.e., volume-outcome relationship)?
Author Response
I thank the reviewer for the complimentary remarks. The items the reviewer is asking to add are addressed n the revised manuscript. Thanks again.